# Surface Characteristics Affect the Properties of PLGA Nanoparticles as Photothermal Agents

**DOI:** 10.3390/mi14081647

**Published:** 2023-08-21

**Authors:** Vera L. Kovalenko, Olga A. Kolesnikova, Maxim P. Nikitin, Victoria O. Shipunova, Elena N. Komedchikova

**Affiliations:** 1Moscow Institute of Physics and Technology, 9 Institutskiy per., 141701 Dolgoprudny, Russia; kovalenko.vl@phystech.edu (V.L.K.); kolesnikova.oa@mipt.ru (O.A.K.); max.nikitin@gmail.com (M.P.N.); elena.komedchikova@gmail.com (E.N.K.); 2Department of Nanobiomedicine, Sirius University of Science and Technology, 1 Olympic Ave., 354340 Sochi, Russia

**Keywords:** PLGA, polymeric nanoparticles, water-in-oil method, photothermal therapy, magnesium(II) phthalocyanine

## Abstract

Photothermal therapy is one of the most promising and rapidly developing fields in modern oncology due to its high efficiency, localized action, and minimal invasiveness. Polymeric nanoparticles (NPs) incorporating low molecular-weight photothermal dyes are capable of delivering therapeutic agents to the tumor site, releasing them in a controlled manner, and providing tumor treatment under external light irradiation. The nanoparticle synthesis components are critically important factors that influence the therapeutically significant characteristics of polymeric NPs. Here, we show the impact of stabilizers and solvents used for synthesis on the properties of PLGA NPs for photothermal therapy. We synthesized PLGA nanocarriers using the microemulsion method and varied the nature of the solvent and the concentration of the stabilizer—namely, chitosan oligosaccharide lactate. A phthalocyanine-based photosensitizer, which absorbs light in the NIR window, was encapsulated in the PLGA NPs. When mQ water was used as a solvent and chitosan oligosaccharide lactate was used at a concentration of 1 g/L, the PLGA NPs exhibited highly promising photothermal properties. The final composite of the nanocarriers demonstrated photoinduced cytotoxicity against EMT6/P cells under NIR laser irradiation in vitro and was suitable for bioimaging.

## 1. Introduction

Photothermal therapy (PTT) has become a promising approach for cancer therapy because it has a number of advantages over traditional methods of treatment—namely, surgery, chemotherapy, and radiotherapy [1]. PTT is a highly selective method that provides a spatially and temporally targeted action on cancer cells, and minimizes the risk of side effects associated with non-specific treatments, ensuring a more focused and effective therapeutic outcome [2,3,4]. Moreover, PTT treatment techniques are capable of inducing the release of tumor-associated antigens through generated heat. This, in turn, leads to the activation of the immune response and the subsequent destruction of cancer cells, resulting in the long-term effectiveness of the therapy [5].

Nanoparticles have significantly revolutionized cancer treatment, offering a multitude of advantages. Through precise targeting and controlled drug release, nanoparticles can minimize off-target effects and reduce systemic toxicity associated with conventional cancer treatments [6]. Encapsulation of photothermal dyes into nanoparticles can enhance the biodistribution of the photothermal dye in the organism and achieve a greater accumulation of therapeutic/diagnostic substances in the tumor in comparison with the introduction of a free dye, which is mainly filtered by the kidneys. Phthalocyanine dyes are particularly notable among photothermal dyes due to their unique physical and chemical properties, such as their light absorption in the transparency window of biological tissues, their thermal stability—allowing for repeated irradiation—and their low absorption in the 400–600 nm range, resulting in reduced sun-induced cytotoxicity [7,8].

PLGA nanoparticles are actively used as a drug encapsulation platform for efficient drug delivery to tumors; this can be attributed to their biocompatibility, controlled drug-release capabilities, and successful approval by the Food and Drug Administration (FDA) for clinical use in diagnostic and therapeutic applications [9,10]. PLGA nanoparticles can be manufactured using reproducible and scalable techniques such as solvent evaporation, the microfluidic method, the solvent displacement method, ionic gelation, and others. One of the most commonly used PLGA synthesis methods is solvent evaporation, allowing the optimization of the properties of nanoparticles by slightly changing the synthesis parameters.

When optimizing nanoparticles for therapeutic and diagnostic applications using the common solvent evaporation synthesis method, most attention is usually paid to the variation of the so-called “main” components of the synthesis, such as the properties of the PLGA polymer—e.g., the molecular weight of the PLGA, the ratio of lactic and glycolic acids, the presence of terminal carboxylic or ester groups, and other parameters [11,12,13]. Much less attention is paid to the influence of those components that are not included in the final composition of the nanoparticle—the solvent used, the pH and ionic strength of buffer systems, the presence of stabilizers, and other parameters. Many research studies have examined how synthesis parameters impact the physicochemical characteristics of PLGA nanoparticles—specifically the size, shape, ζ-potential, and also their cytotoxic properties in vitro [14,15,16,17,18]. However, the choice of coating and stabilizer solvent has a significant impact on the nanoparticle size and therapeutic molecule encapsulation efficiency.

In this work, we investigated for the first time the role of the stabilizer solvent and coating concentration during PLGA synthesis in modulating the photothermal properties of phthalocyanine-loaded PLGA NP in vitro and ex vivo bioimaging studies.

## 2. Materials and Methods

### 2.1. Synthesis of Polymeric Nanoparticles

PLGA was used as a matrix for nanoparticle synthesis by the oil-in-water microemulsion method, as described by us previously [19]. Briefly, the emulsion was formed by adding 350 µL of a dichloromethane solution containing 40 g/L PLGA and 1 g/L of magnesium(II) phthalocyanine (Sigma, Darmstadt, Germany) to 3 mL of 3% aqueous polyvinyl alcohol, PVA (Mowiol 4-88, Sigma, Darmstadt, Germany) with 1 g/L chitosan oligosaccharide lactate (5 kDa, Sigma, Darmstadt, Germany). The emulsion was sonicated for 1 min at 35% amplitude and 200 W power and gently shaken for 40 min for dichloromethane evaporation. The obtained nanoparticles were washed by triple centrifugation for 10 min at 2500× *g* and resuspended in 200 µL of Milli-Q water (mQ). The final concentration of the nanoparticles was determined by drying at 90 °C.

### 2.2. Electron Microscopy

Images of synthesized nanoparticles were obtained with an MAIA3 scanning electron microscope Tescan (Tescan, Brno, Czech Republic) at an accelerating voltage of 7 kV. Samples of nanoparticles resuspended in water at a concentration of 10 µg/L were air-dried on a silicon wafer on a carbon film and analyzed immediately. SEM images were processed using ImageJ software to get the size distribution of the nanoparticles.

### 2.3. Photothermal Properties Study

A colloidal solution of nanoparticles at a concentration of 1 g/L in phosphate-buffered saline (PBS) was irradiated with a 1.2 W/cm^2^ 808 nm laser in a 2 mL Eppendorf-like tube in triplicate. The temperature change was recorded continuously with an FLIR One Pro thermal imaging camera (Teledyne FLIR, Santa Barbara, CA, USA). The experiment was conducted at a constant room temperature. The samples were in a thermostat at 27 °C before irradiation to eliminate temperature differences between samples.

### 2.4. ζ-Potential Measurements

The ζ-potential of the PLGA nanoparticles was measured using the electrophoretic light scattering technique with a ZetaSizer Nano ZS analyzer (Malvern Instruments Ltd., Worcestershire, UK) in 10 mM NaCl.

### 2.5. Measurement of Pht-Mg Incorporation

The incorporation of Pht-Mg was investigated by fluorescence spectroscopy. PLGA nanoparticles were dissolved in a DMSO:mQ ratio of 1:1 and then fluorescence was measured using a CLARIOstar Plus microplate analyzer (BMG LABTECH) at λ_ex_ = 669 nm, λ_em_ = 700 nm. The incorporation of Pht-Mg was calculated using a fluorescence calibration curve for Pht-Mg in the same solution.

### 2.6. Cell Culture

EMT6/P mouse mammary cells (BRC Lab collection, MIPT, Moscow, Russia) were cultivated in DMEM medium (PanEco, Moscow, Russia) supplemented with 10% fetal bovine serum (HyClone, Logan, UT, USA), 2 mM L-glutamine (PanEco, Moscow, Russia), and penicillin/streptomycin (PanEco, Moscow, Russia) and incubated in a humidified atmosphere with 5% CO_2_ at 37 °C.

### 2.7. Flow Cytometry Analysis

To determine the specificity of the obtained nanoparticles, 1 × 10^6^ EMT6/P cells were resuspended in 300 μL PBS with 1% BSA and mixed with PLGA nanoparticles at various concentrations. The mixture was incubated for 15 min, then cells were washed from unbound nanoparticles by centrifugation at 100× *g* and analyzed on a Novocyte 2000R flow cytometer (ACEA Biosciences, San Diego, CA, USA; excitation laser 640 nm, emission filter 675/30 nm).

### 2.8. Cytotoxicity Assay

The cytotoxicity of the nanoparticles was determined using a resazurin-based toxicity assay. Cells were incubated with nanoparticles at various concentrations in 200 μL complete DMEM medium without phenol red in a 2 mL tube for 2 h in a humidified atmosphere with 5% CO_2_ at 37 °C, and then irradiated with an 808 nm laser. The cells were then diluted with full medium and seeded on a 96-well plate at 2 × 10^3^ cells per well in 150 μL DMEM medium with 10% FBS and cultured for 48 h. After 48 h of incubation, the medium was removed and 100 μL of resazurin solution (13 mg/L in PBS) was added to each well. The samples were incubated for 1.5 h at 37 °C in a humidified atmosphere with 5% CO_2_. Fluorescence was measured using a CLARIOstar Plus microplate analyzer (BMG LABTECH) at λ_ex_ = 570 nm, λ_em_ = 600 nm. Data are presented as a percentage of untreated cells.

### 2.9. Bioimaging Ex Vivo

To measure the bioimaging abilities of the nanoparticles, empty glass capillaries and capillaries with 40 µg of nanoparticles were covered with boneless turkey thigh fillet slices, and the fluorescence intensity was measured. Boneless turkey thigh fillets were cut into slices from 1 to 10 mm thick to obtain different thicknesses of tissue. Two groups of glass capillaries (*n* = 3 in each group) were used: control empty capillaries and capillaries with the addition of 10 µL of PLGA NP at a concentration of 8 g/L. Ex vivo imaging was performed with a LumoTrace FLUO bioimaging system (Abisense, Sochi, Russia) as follows: the meat was imaged with fluorescence at λ_ex_ = 730 nm and a 780 nm long-pass filter.

## 3. Results

### 3.1. Design of the Experiment

The design of the study is presented in Figure 1 and included two parts:(i)The selection of the optimal stabilizer solvent and coating concentration during PLGA nanoparticle synthesis. Phthalocyanine-loaded PLGA nanoparticles were synthesized using the microemulsion solvent evaporation method. Polyvinyl alcohol (PVA) was chosen as the stabilizer for the NP synthesis as it is the most frequently used stabilizing agent [20]. The chitosan oligosaccharide lactate was used as a coating due to its ability to reduce the burst release of the encapsulated drugs. To obtain PLGA NP with the most prominent photothermal properties, the PVA solvent and the concentration of the chitosan lactate were varied during NP synthesis.(ii)The optimal PLGA formulation was chosen to further test NIR-induced cytotoxicity on cancer cells in vitro and ex vivo bioimaging studies.

### 3.2. Effect of Stabilizer Solvent during Synthesis on the Photothermal Properties of PLGA NP

First, we studied the influence of the PVA solvent used during synthesis on the photothermal properties of the PLGA NP. Phthalocyanine-loaded PLGA NPs were synthesized using Milli-Q-water (mQ), pH 6.2, sodium acetate buffer, pH 4.5 (CH_3_COONa) or PBS buffer, pH 7.4 as the PVA solvent (Figure 2a). The concentration of chitosan lactate was maintained at 1 g/L in all these buffer systems. The photothermal properties of the magnesium phthalocyanine-loaded PLGA NPs (PLGA/Pht-Mg) were investigated by irradiating a solution of NPs in water with an 808 nm laser for 10 min.

PLGA nanoparticles synthesized with mQ and PBS as solvents had superior photothermal properties compared to PLGA synthesized with sodium acetate as the solvent (Figure 2b). These two types of nanoparticles reached a plateau after 7 min of irradiation; however, NPs prepared with mQ had a higher heating rate and were selected as the PVA solvent for further syntheses.

The measurement of the Pht-Mg loading efficiency showed that the dye loading was equal to 46.3 ± 6.2, 30.2 ± 8.2, and 16.0 ± 2.4 µg of Pht-Mg per 1 mg of PLGA/Pht-Mg for nanoparticles synthesized with mQ, PBS, and sodium acetate, respectively—thus demonstrating an 80.4%, 61.7%, and 49.9% loading efficiency during the synthesis (225 µg of Pht-Mg were used in the synthesis per 36 mg of PLGA).

### 3.3. Effect of Coating Concentration during Synthesis on the Physicochemical Characteristics of PLGA NPs

Next, to study the effects of the coating concentration—namely, of the chitosan oligosaccharide lactate—on the properties of PLGA nanoparticles, a series of syntheses of PLGA/Pht-Mg NP were carried out as shown in Figure 3a. The chitosan lactate concentration during the synthesis was varied from 0 g/L to 3 g/L. Scanning electron microscopy (SEM) was used to obtain micrographs of PLGA/Pht-Mg (Figure 3b), which were then processed using the ImageJ software package (Figure 3c). According to the obtained data (Figure 3d), it was found that there was no direct correlation between the size of the NPs and the chitosan lactate concentration during the synthesis.

The ζ-potential of PLGA nanoparticles with different concentrations of chitosan lactate was studied using the electrophoretic light scattering method; the results are presented in Table 1 and Figure A1. From the obtained data, it is difficult to draw an unambiguous conclusion about the influence of chitosan lactate concentration on the ζ-potential of PLGA/Pht-Mg NPs.

Then, the effects of the chitosan lactate concentration on the photothermal properties of the nanoparticles were investigated. Resuspended in mQ, nanoparticles were irradiated with an 808 nm laser for 10 min. As shown in Figure 3e, nanoparticles with a concentration of chitosan lactate at 3 g/L during the synthesis had the most pronounced photothermal properties—namely, the nanoparticles heated up to 40 °C in 10 min.

The measurement of the Pht-Mg loading efficiency showed that the dye loading was equal to 28.3 ± 0.2, 25.8 ± 0.3, 55.6 ± 6.2, 46.3 ± 0.7, and 72.7 ± 6.7 µg of Pht-Mg per 1 mg of PLGA/Pht-Mg for nanoparticles synthesized with 0, 0.083, 0.5, 1, and 3 g/L of chitosan lactate, respectively—thus demonstrating a 37.7%, 34.3%, 74.1%, 61.7%, and 96.9% loading efficiency during the synthesis (225 µg of Pht-Mg were used during the synthesis per 36 mg of PLGA). Chitosan lactate is known to reduce the leakage of the encapsulated drug, which results in an enhanced loading efficiency as well as Pht-Mg retention inside PLGA nanoparticles [21,22].

Based on the obtained data, by varying the concentration of chitosan lactate, it is possible to influence the photothermal properties of nanoparticles without affecting their physicochemical properties—namely, their size and ζ-potential.

### 3.4. In Vitro Study of PLGA/Pht-Mg Interaction with Cells: Flow Cytometry Analysis and In Vitro Cytotoxicity Tests

To investigate the non-specific binding of the synthesized nanoparticles, PLGA NPs were incubated with EMT6/P cells and the binding efficiency was measured using flow cytometry (Figure 4a). Due to the fact that cancer cells have a negative surface charge, an increase in the concentration of positively charged chitosan on the surface of nanoparticles led to an obvious increase in the binding of nanoparticles to cells, due to electrostatic interactions. This trend was observed up to a concentration of 1.5 g/L. However, upon reaching a concentration of 3 g/L, a significant decrease in the binding of nanoparticles to cells was observed. Most likely, this occurred due to the complete coverage of the surface of nanoparticles with chitosan lactate, which led to an increase in the aggregation and sedimentation stability of the nanoparticles—which, in turn, led to reduced non-specific binding.

For further experiments, we selected nanoparticles with a concentration of 3 g/L of chitosan lactate during the synthesis as they had the most prominent photothermal properties and decreased non-specific binding. In order to confirm the photoinduced cytotoxicity of the synthesized nanoparticles, a series of cytotoxicity studies were carried out on EMT6/P cells. Cells were incubated with polymeric nanoparticles at varying concentrations for 2 h, followed by the removal of unbound nanoparticles. Then, the cell suspension was subjected to 808 nm irradiation. After 48 h of cultivation, a resazurin-based cytotoxicity test was performed (Figure 4b). As a result, the nanoparticles demonstrated an absence of dark cytotoxicity (cytotoxicity without laser irradiation), while the laser-induced cytotoxicity of the tested nanoparticles was concentration-dependent, with the highest cytotoxicity observed at a concentration of 1 g/L.

### 3.5. Bioimaging Study of PLGA/Pht-Mg NP Ex Vivo

PLGA/Pht-Mg nanoparticles synthesized with 3 g/L of chitosan lactate were placed in glass capillaries and covered with fillets of different thicknesses to simulate different tissue depths. As shown in Figure 5, the fluorescence of nanoparticles can be accurately detected at a depth of up to 3 mm. With an increase in tissue thickness, the fluorescence from the particles remained detectable even at a depth of 7 mm, although their precise localization could not be determined. This suggests that the nanoparticles have the potential to provide diagnostic information, but further improvements may be necessary to precisely identify their exact location within thicker tissues.

## 4. Discussion

Nanomedicine has emerged as a promising field for cancer treatment, offering innovative approaches for improving traditional methods of treatment. In particular, oncotheranostics is of great interest due to its focus on the development of nanoagents that possess both diagnostic and therapeutic capabilities simultaneously. Oncotheranostic nanoagents can be designed to specifically target cancer cells, delivering therapeutic agents directly to the tumor site with high precision—reducing side effects and improving treatment outcomes [23].

Polymeric particles, specifically PLGA, are receiving significant attention among oncotheranostic agents due to their biocompatibility, non-toxic nature, and controlled drug-release properties [24]. PLGA nanoparticles can encapsulate a wide range of therapeutic agents, including drugs and imaging agents, protecting them from degradation and improving their stability and biodistribution [25,26,27]. Moreover, PLGA NPs can be easily customized through slight modifications in their synthesis parameters, which affect their size, surface charge, and composition.

PLGA NPs have applications across various fields of cancer treatment [19,26,27,28,29]—in particular, PLGA NPs are utilized as carriers in photothermal therapy. Photothermal therapy (PTT) is a non-invasive and highly selective method. PTT is based on the ability of photosensitizers—namely, nanoparticles or dyes—to convert absorbed light into heat. This heat generation leads to localized hyperthermia within the targeted tissue or tumor, resulting in thermal ablation or damage to cancer cells. By controlling the laser parameters, such as the power and exposure time, the intensity of the heat produced can be precisely regulated.

The selection of a photosensitizer is a critical factor in determining the efficacy of phototherapy. Phthalocyanine dyes play a significant role in photothermal therapy due to their unique characteristics. These dyes exhibit strong light absorption within the near-infrared region, which enables deep tissue penetration and selective heating of the target area upon laser irradiation. Moreover, phthalocyanine dyes are known for their low cytotoxicity in the visible light range and their thermal stability.

The achievement of optimal outcomes in phototherapy relies on the efficient accumulation of photosensitizers within the tumor. This crucial aspect is highly dependent on the characteristics of the nanoparticles, which can be precisely fine-tuned by adjusting the synthesis parameters.

We selected the microemulsion solvent evaporation method as a commonly employed technique for PLGA NP synthesis, which has demonstrated the successful incorporation of substances with diverse natures [29,30,31,32,33,34,35,36,37]. The role of the solvent in the synthesis of PLGA nanoparticles is crucial, as it significantly impacts the efficiency of droplet formation during the emulsification process, the stability of droplets, and affects drug encapsulation. While many studies have primarily focused on investigating the influence of the polymer solvent on the characteristics of PLGA nanoparticles, it is important to acknowledge the contribution of the stabilizer solvent as well.

PVA is a commonly used stabilizer in the preparation of PLGA nanoparticles, affecting the size of the PLGA nanoparticles, their colloidal stability, and their drug encapsulation efficiency [38,39]. Here, we studied three commonly used buffer systems—namely, mQ water, PBS, and sodium acetate—as PVA solvents. The osmolarity and pH of the solvent can affect the efficiency of the incorporation of various substances into PLGA nanoparticles—in particular, photosensitizers. Our results demonstrate that, for the efficient encapsulation of hydrophobic dyes, mQ water emerges as the optimal PVA solvent. However, it is important to note that these findings may vary when considering hydrophilic substances.

The coating of PLGA nanoparticles is another significant synthesis parameter that plays a crucial role in determining nanoparticle size, preventing the leakage of encapsulated substances, and impacts the interactions between nanoparticles and cells. In our study, we specifically chose chitosan oligosaccharide lactate as a coating for the PLGA nanoparticles due to its numerous advantages; these include compatibility with biological systems, extended circulation time of the PLGA nanoparticles, minimized burst release of encapsulated substances, controlled drug delivery, and enhanced cellular uptake owing to its positive charge. In our study, we examined a broad range of chitosan lactate concentrations to identify the most optimal for the efficient incorporation of the dye and its pronounced photothermal properties. Interestingly, the concentration of chitosan lactate did not significantly impact the physicochemical properties of the particles, including their size and zeta potential. This suggests that by varying the coating concentration, we can enhance the photothermal properties of the nanoparticles and modulate their interactions with cells without introducing notable changes to their physicochemical characteristics.

In this study, we demonstrated the effects of PVA solvent on the photothermal properties of PLGA nanoparticles. In addition, we investigated the relationship between the concentration of chitosan lactate used during the synthesis and the size, charge, and photothermal properties of the obtained nanoparticles. The results showed that the PVA solvent and chitosan lactate concentration did not directly affect the size and charge of the nanoparticles, but did influence their photothermal properties. MQ as a PVA solvent and a chitosan lactate concentration of 3 g/L were chosen for the synthesis to obtain the optimal nanoparticle formulation for successful photoinduced cytotoxicity in vitro and bioimaging ex vivo. These findings emphasize the importance of conducting further investigations to gain a deeper understanding of the complex relationships between the synthesis factors and resulting nanoparticles. By refining our understanding, we can optimize the design and synthesis of nanoparticles for enhanced photothermal applications.

## 5. Conclusions

In conclusion, our study examined the role of the PVA solvent and chitosan lactate concentration used during synthesis on the size, charge, and photothermal properties of nanoparticles. Our results indicated that the PVA solvent and chitosan lactate concentration did not directly affect the physico-chemical properties of the PLGA NPs, but had a significant influence on their photothermal properties.

Overall, our study highlights the importance of considering the role of solvents and synthesis parameters on the photothermal properties of nanoparticles, providing valuable insights for the development of advanced nanomaterials in the field of photothermal therapy. Continued investigations in this area will deepen our knowledge and contribute to the advancement of nanoparticle-based therapeutic approaches for various biomedical applications.

## Figures and Tables

**Figure 1 micromachines-14-01647-f001:**
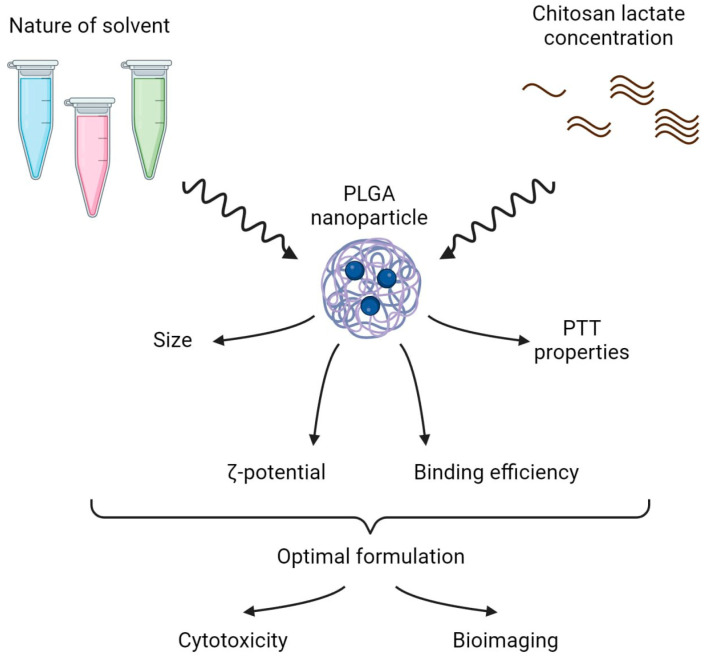
Stabilizer solvent and coating concentration screening—scheme of the experiment. In the first step, solvents of different natures and various chitosan lactate concentrations were tested to obtain the PLGA nanoformulation with the optimal physicochemical and photothermal properties. In the second step, dark and photoinduced cytotoxicity in vitro and ex vivo bioimaging of the optimal nanoformulations were studied.

**Figure 2 micromachines-14-01647-f002:**
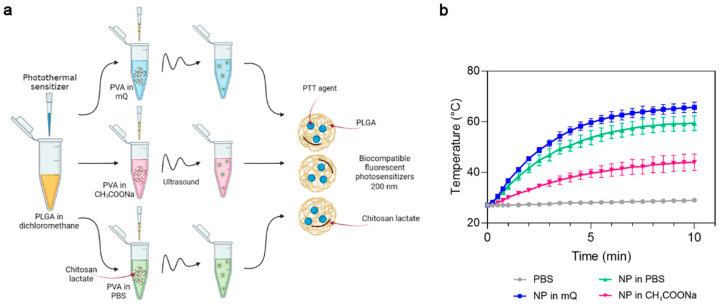
The choice of optimal PVA solvent in the synthesis of PLGA/Pht-Mg nanoparticles. (**a**) Scheme of PLGA nanoparticle synthesis. A mixture of PLGA and Pht-Mg in dichloromethane was poured into a PVA solution containing chitosan lactate oligosaccharide at different concentrations and sonicated. (**b**) Photothermal properties upon 808 nm laser irradiation of PLGA/Pht-Mg NP synthesized using PVA dissolved in water (mQ), pH 6.2, acetate buffer, pH 4.5 or PBS, pH 7.4.

**Figure 3 micromachines-14-01647-f003:**
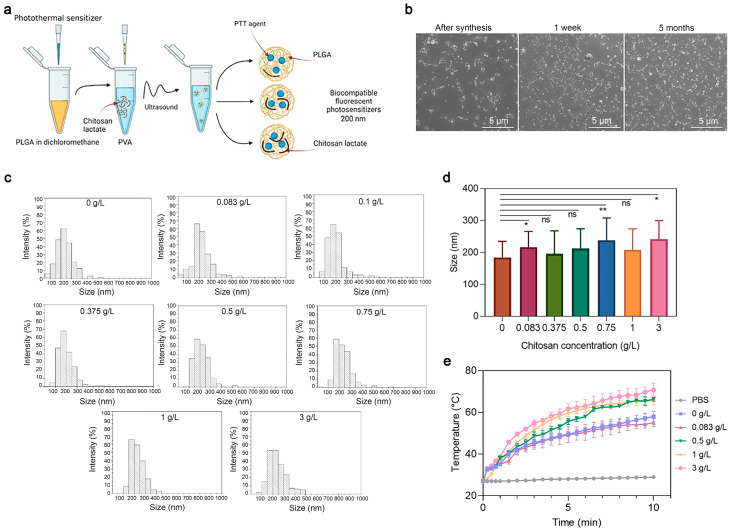
Synthesis and characterization of PLGA nanoparticles. (**a**) Scheme of PLGA nanoparticle synthesis. A mixture of PLGA and Pht-Mg in dichloromethane was poured into the PVA solution containing chitosan lactate oligosaccharide at different concentrations and sonicated. (**b**) Scanning electron microscopy images of PLGA/Pht-Mg NP obtained right after synthesis, 1 week after and 5 months after synthesis. (**c**) Size distribution of PLGA/Pht-Mg nanoparticles with various chitosan lactate concentrations used during the synthesis, obtained via SEM image processing. (**d**) The average size of nanoparticles with various chitosan lactate concentrations used during the synthesis, obtained via SEM image processing. Data presented are mean ± s.d. * *p* < 0.05, ** *p* < 0.01, ns *p* ≥ 0.05, Paired two-sample t-test for mean. (**e**) Photothermal properties of PLGA/Pht-Mg nanoparticles synthesized with chitosan lactate at the concentrations 0 g/L; 0.08 g/L; 0.5 g/L; 1 g/L; and 3 g/L.

**Figure 4 micromachines-14-01647-f004:**
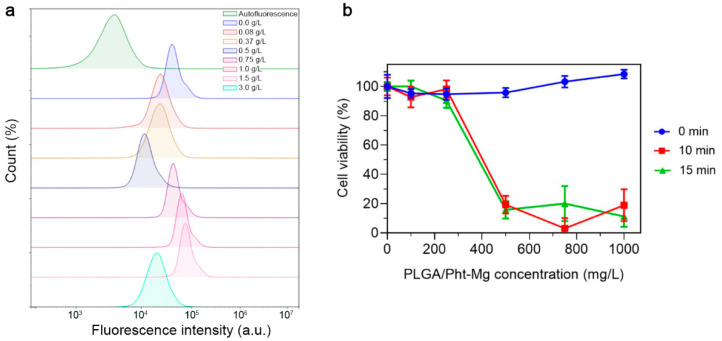
Study of PLGA/Pht-Mg NP interactions with EMT6/P cells. (**a**) Flow cytometry analysis of PLGA/Pht-Mg NP binding efficiency to EMT6/P cells. (**b**) Cytotoxicity of PLGA/Pht-Mg NP under 808 nm laser light irradiation, measured by a resazurin test.

**Figure 5 micromachines-14-01647-f005:**
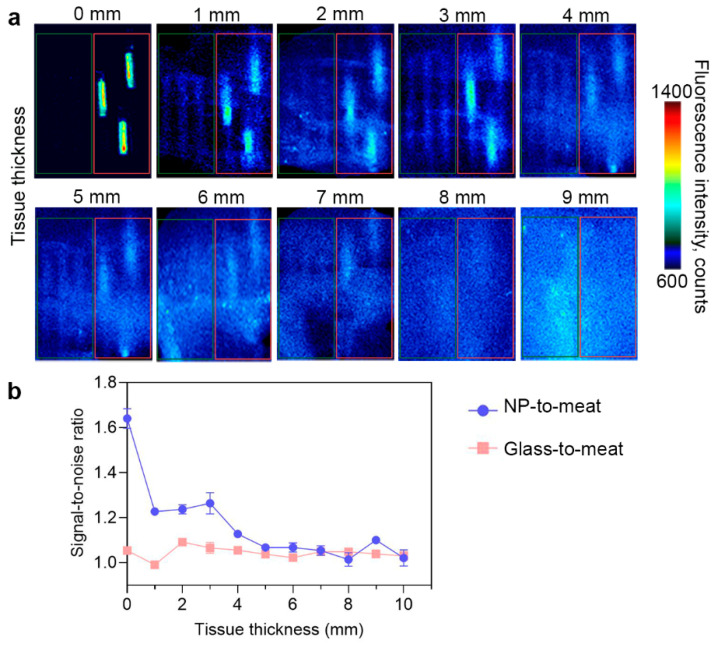
Bioimaging study of PLGA/Pht-Mg NP ex vivo. (**a**) Fluorescent images of empty glass capillaries (shown in the red frame) and glass capillaries with PLGA/Pht-Mg NP (shown in the green frame) covered with turkey fillet. (**b**) Fluorescence intensity of PLGA/Pht-Mg NP at different tissue depths. Dots represent the signal-to-noise ratio of the fluorescence intensity from capillaries with PLGA NPs or empty glass capillaries to meat.

**Table 1 micromachines-14-01647-t001:** Zeta potential study of PLGA/Pht-Mg nanoparticles with several concentrations of chitosan oligosaccharide lactate.

Chitosan Concentration (g/L)	Zeta Potential (mV)
0	–2.0 ± 4.0
0.08	–5.3 ± 5.4
0.38	–5.8 ± 3.7
0.5	–4.4 ± 3.9
0.75	–5.2 ± 4.3
1.5	–5.2 ± 3.8
3.0	–1.8 ± 3.5

## Data Availability

All data are presented within the manuscript.

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
