# Peer review of "Surface Characteristics Affect the Properties of PLGA Nanoparticles as Photothermal Agents"

_micromachines, 2023, doi:10.3390/mi14081647_

Round 1
Reviewer 1 Report
Cancer is one of the most challenging diseases currently worked on by various researchers and a variety of nanomedicine-based approaches are being explored. Therapeutic approached like hyperthermia, photodynamic therapy and photothermal therapy are continually attempted to be improved upon.
In this manuscript, the authors have attempted to improve on the physico-chemical properties PLGA nanoparticles utilized for photothermal therapy for enhanced therapeutic efficacy. Their findings are quite encouraging and the manuscript is an interesting read. However, before being considered for publication, the manuscript requires attention towards addressing the following issues.
1. The synthesis of PLGA nanoparticles with optimal phyisco-chemical properties have been well documented over last 20 years. To name a few:
Journal of Biomaterials Applications. 2013;27(7):909-922
Future Sci OA. 2018 Feb; 4(2): FSO263
Mater. Adv., 2022, 3, 837-858
Pharmaceutics 2022, 14(3), 638
Pharmaceutics. 2022, 14(4): 870
How does this work add new information in our understanding of optimizing PLGA nanoparticles?
2. Page 2, Section 2.1: While the authors intend to study the effects of external parameters on the fabrication of PLGA nanoparticles, why did they chose to evaluate the effects only on the dye-loaded ones and not the bare/empty ones? How is the presence of dye affecting the synthesis of high quality PLGA particles?
3. Page 4, Section 3.2: while studying the influence of solvents, only 2 solvents of 2 different pHs were used. Why did the authors not try other buffers with a varied pH range? The current information only shows the effect of neutral pH against acidic pH on the final nanoparticles. What was the pH of MQ water used as a solvent?
4. In continuation to pt 3, Figure 2b shows the time dependent temperature increase of the prepared nanoparticles. At a glance those 3 curves have quite close performance having no significant difference. The samples from MQ water and PBS are saturating and overlapping after 7 min. There is no evidence of multiplicates of this study as the error bars are not visible in the graph. It is suggested to perform this study atleast thrice and add error bars to the graph for a more comprehensive and reliable information.
5. How much nanoparticles/ concentration was used for the study depicted in Fig 2b? As magnesium(II) phthalocyanine is solely responsible for the heating, what was its loading efficiency? Were the amount of particles normalized to the dye content before evaluating their heating performances?
6. Fig 3e, although 1g/l of sample show the lowest temperature rise, both 0 g/l and 3g/l of chitosan lactate do not show any significant different in their heating performances. The error bars should be included in the graph to provide better understanding.
7. In both Fig 2b and 3e although the temperature rise is studied until saturated which is 60-70 ° C, the temperature needed to initiate apoptosis in cancer cells is below 45 °C. All the variety of prepared samples show to reach that temperature within 2 min which indicates that changing these parameters may or may not have the desired effect on the final PLGA nanoparticles.
For both the figures the PLGA nanoparticles used for the study should be normalized to either the amount of dye or to the amount of PLGA.
8. For the evaluation of non-specific interactions with the cell surface, fig 4a shows a trend. As the concentration of chitosan lactate is increased there is a decrease in non-specific binding until 0.5 g/l which then starts to increase until 1.5 g/l and suddenly drops back for 3 g/l sample. What do the authors think is the reason behind this behaviour and chitosan-dependent binding?
9. The tick labels of x-axis should be more inclusive and not just 2 points, to give better understanding of the variety of concentrations used for evaluation.
10. Page 7, section 3.4: Figure is wrongly labelled as fig6 in text. It should be figure 5.
After thorough reading and careful consideration of this work, it is advised to reconsider the manuscript after addressing the comments and undertaking the above suggested revisions.
Minor edits required
Author Response
Dear reviewer, please find attached pdf file with pont-by-point responses.

Reviewer 2 Report
Review for “The Surface Characteristics Affect the Properties of PLGA Nanoparticles as Photothermal Agents” by Kovalenko et al.
This manuscript investigates the impact of stabilizers and solvents in the synthesis of PLGA nanoparticle therapeutic carriers. The main goal of this paper is to study the oncology and imaging potential of photothermal dyes encapsulated PLGA particles and the impact of stabilizers and solvents on their performance. The abstract and introduction of the paper are straightforward and well-written. My biggest concerns about this manuscript are the design of the experiment, the characterization of the nanoparticles, and some data analysis. This study is publishable with major revision.
In section 3.3, the authors should have a negative control of cell viability where cells are not labeled with the PLGA particles but are subject to the laser of 10/15 min. Without this information, it remains unclear whether the cytotoxicity is from laser exposure itself.
In section 3.2 (second), the size of the nanoparticles was evaluated using SEM in a dry environment. This can create large deviations from sizes measured in liquid. For this paper, DLS measurements of nanoparticle hydrodynamic sizes should be used. The reviewer suspects hydrodynamic sizes of nanoparticles obtained from DLS will be significantly different from SEM and may reveal some variations among particles prepared from different coating concentrations. In addition, the authors should also provide stability of the nanoparticles with respect to time (say a week) in various biological solvents used in the study. Here both DLS size and fluorescence signal should be presented. This is related to concerns of particle aggregation with respect to time due to the large sizes of particles. And to address concerns about the uncontrolled release of dye during storage.
In section 3.4, fluorescence images of the tracers under different tissue depths are presented. However, there is no detailed data analysis method described in the method section. How is the intensity percentage calculated in Figure 5? The authors should at least describe the signal-to-noise ratio, how the percentage is calculated, and if possible, perform a control group where the thickness of tissue is the same but without particles. Another concern with the design of the imaging experiment is that the particles are injected into the tissue and the images were taken tens of minutes after, there could be particle migration between tissue thicknesses. This can cause errors in thickness identification. A better way would be to have two groups of experiments. One group with controls, only tissue, and an empty tube, varying tissue thickness. Another group with particles in the tube, varying tissue thickness.
There are also minor concerns on:
1. In the introduction, paragraph 2, the authors state that “Encapsulation of photothermal dyes into nanoparticles can enhance the biodistribution of the nanoparticles in the organism and achieve a greater accumulation of therapeutic/diagnostic substances in the tumor”. There is no supporting literature on the statement. And the reviewer has strong objections to the statement. Certain nanocarriers do can preferentially accumulate in the tumor, this is either due to external forces (such as magnetic force) or due to the size of nanocarriers (such as large enough to be caught up by macrophages and goes to organs like the liver, and liver must be the target organ). However, it is unclear to the reviewer why the encapsulation of photothermal dyes into the nanoparticles can modulate biodistribution and achieve greater accumulation.
2. The result section has two section 3.2
3. Was the photothermal data calibrated by room temperature solvent at the same time to eliminate the effect of environmental temperature change for different measurements? Or is the experiment conducted in an environment where the temperature is constant? This should be clarified in the experimental section.
4. The authors should also discuss in section 3.2 (second) why different coating concentrations can lead to different photothermal properties of the particles. Is this related to encapsulation efficiency? The authors should also consider adding characterization on encapsulation efficiency in different combinations of stabilizer and coating concentrations.
English fine, minor edits.
Author Response

(The authors gave the same response as above.)

Round 2
Reviewer 1 Report
Can be accepted in its revised format
Needs minor edits
Reviewer 2 Report
The manuscript is recommended to accept in current form.